# Cystic Fibrosis Defective Response to Infection Involves Autophagy and Lipid Metabolism

**DOI:** 10.3390/cells9081845

**Published:** 2020-08-06

**Authors:** Alessandra Mingione, Emerenziana Ottaviano, Matteo Barcella, Ivan Merelli, Lorenzo Rosso, Tatiana Armeni, Natalia Cirilli, Riccardo Ghidoni, Elisa Borghi, Paola Signorelli

**Affiliations:** 1Biochemistry and Molecular Biology Laboratory, Health Science Department, University of Milan, San Paolo Hospital, 20142 Milan, Italy; alessandra.mingione@unimi.it (A.M.); riccardo.ghidoni@unimi.it (R.G.); 2Laboratory of Clinical Microbiology, Health Science Department, University of Milan, San Paolo Hospital, 20142 Milan, Italy; emerenziana.ottaviano@unimi.it (E.O.); matteobarcella84@gmail.com (M.B.); elisa.borghi@unimi.it (E.B.); 3National Research Council of Italy, Institute for Biomedical Technologies, Via Fratelli Cervi 93, 20090 Milan, Italy; ivan.merelli@itb.cnr.it; 4Health Sciences Department, University of Milan, Thoracic surgery and transplantation Unit, Fondazione IRCCS Ca Granda Ospedale Maggiore Policlinico, 20122 Milan, Italy; lorenzo.rosso@unimi.it; 5Department of Clinical Sciences, Section of Biochemistry, Biology and Physics, Polytechnic University of Marche, 60131 Ancona, Italy; t.armeni@staff.univpm.it; 6Cystic Fibrosis Referral Care Center, Mother-Child Department, United Hospitals Le Torrette, 60126 Ancona, Italy; natalia.cirilli@ospedaliriuniti.marche.it; 7“Aldo Ravelli” Center for Neurotechnology and Experimental Brain Therapeutics, via Antonio di Rudinì 8, 20142 Milan, Italy

**Keywords:** cystic fibrosis, sphingolipids, autophagy, myriocin, *Aspergillus fumigatus*

## Abstract

Cystic fibrosis (CF) is a hereditary disease, with 70% of patients developing a proteinopathy related to the deletion of phenylalanine 508. CF is associated with multiple organ dysfunction, chronic inflammation, and recurrent lung infections. CF is characterized by defective autophagy, lipid metabolism, and immune response. Intracellular lipid accumulation favors microbial infection, and autophagy deficiency impairs internalized pathogen clearance. Myriocin, an inhibitor of sphingolipid synthesis, significantly reduces inflammation, promotes microbial clearance in the lungs, and induces autophagy and lipid oxidation. RNA-seq was performed in *Aspergillusfumigatus*-infected and myriocin-treated CF patients’ derived monocytes and in a CF bronchial epithelial cell line. Fungal clearance was also evaluated in CF monocytes. Myriocin enhanced CF patients’ monocytes killing of *A. fumigatus*. CF patients’ monocytes and cell line responded to infection with a profound transcriptional change; myriocin regulates genes that are involved in inflammation, autophagy, lipid storage, and metabolism, including histones and heat shock proteins whose activity is related to the response to infection. We conclude that the regulation of sphingolipid synthesis induces a metabolism drift by promoting autophagy and lipid consumption. This process is driven by a transcriptional program that corrects part of the differences between CF and control samples, therefore ameliorating the infection response and pathogen clearance in the CF cell line and in CF peripheral blood monocytes.

## 1. Introduction

Cystic fibrosis (CF) is a hereditary disease associated with different classes of mutations in the Cystic Fibrosis Transmembrane conductance Regulator (CFTR), a chloride/carbonate channel. A deletion of phenylalanine 508 (F508) affects more than 70% of patients and results in unfoldedproteins accumulation, causing a proteinopathy responsible for inflammation, impaired autophagy, and altered lipid metabolism [1,2,3,4,5]. CF is a multiple-organ disease characterized by life-threatening chronic inflammation and recurrent infection of the lungs, where defective immune response and altered mucus viscosity and acidity pave the way to persistent microbial colonization [6]. Indeed, CF is characterized by ineffective clearance of pathogens and reduced killing of internalized microbes, both in pulmonary airways epithelia and in macrophages [7,8,9]. Moreover, recurrent antibacterial therapies favor the insurgence of drug resistance and, by altering the microbiological milieu, promote colonization by opportunistic pathogens. *Aspergillus fumigatus* is the most prevalent filamentous fungus in the respiratory tract of CF patients, contributing to lung deterioration. Approximately 35% of CF patients are infected by *A. fumigatus*, and researchers are still investigating ways to prevent its colonization [10], since CFTR dysfunction itself has been directly associated with a reduced clearance ability of *A. fumigatus* conidia by CF airway epithelia [8,11].

CF cells suffer from defective autophagy, which is related to altered proteostasis and chronic inflammation [12,13,14]. Autophagy is a conserved cell-autonomous stress response, dedicated to the breakdown of cellular material and to cell content recycling. Apart from its homeostatic role and its crucial activity in stress conditions, autophagy is actively involved in pathogen clearance, offering the cell an extremely efficient defensive response. A specialized form of autophagy called xenophagy involves the recognition and clearance of foreign particles and pathogens. Pattern recognition receptors (PRRs), upon antigen engagement, recruit microtubule-associated protein 1B light chain-3 (LC3) and successively other components of the canonical autophagy pathway, therefore enabling the binding of foreign particles that are already contained within a single membraned phagosome or endosome and their addressing to lysosomes [15].

The ability to detoxify and/or assimilate host lipids is a crucial aspect of the infection process [16,17], particularly in fungal infection [18]. In addition, the mobilization of internal lipid stores to release growth substrates, as well as lipid-mediated cellular signaling, contribute to the pathogen invasion outcome [18]. The activation of lipid catabolism via transcriptional activity of transcription factor EB/peroxisome proliferator-activated receptor α (TFEB/PPARα) potentiates macrophage response to infection [19].

Several studies reported an altered lipid homeostasis in blood and peripheral tissues in CF patients [1,2,20,21,22,23,24], and demonstrated the pathological role of the inflammatory ceramide accumulation in CF airways [4,25]. In a CF mouse model and in bronchial epithelial cells, our group previously demonstrated that hampering ceramide accumulation by the sphingolipid synthesis inhibitor myriocin (Myr) reduces inflammation and ameliorates the response against microbial infection in vivo and in vitro by promoting cell-killing ability [8,25,26]. Moreover, the inhibition of ceramide synthesis in CF cells activates TFEB, a master regulator of the stress response. Indeed, TFEB is responsible for enhancing autophagy and lipid oxidation via the activation of lipid-metabolism-related transcription factors such as PPARs and FOXO, thus reducing the overall lipid content [5].

Here we studied the myriocin modulation of gene expression profile and its effect in promoting conidial killing in CF patients’ monocytes infected with *A. fumigatus*. Next, we demonstrated that *A. fumigatus* infection triggers a different expression profile in CF bronchial epithelial cells compared to healthy control cells. We further investigated the effect of Myr treatment on the expression of genes involved in inflammation and in the autophagic response to infection. Our data suggest that regulating the altered lipid metabolism could represent a possible therapeutic strategy in ameliorating CF disease.

## 2. Materials and Methods

### 2.1. Cells and Treatments

IB3-1 cells (named CF cells), an adeno-associated virus-transformed human bronchial epithelial cell line derived from a CF patient (ΔF508/W1282X) and provided by LGC Promochem (US), were grown in LHC-8 medium supplemented with 10% fetal bovine serum (FBS) and 1% penicillin/streptomycin at 37 °C and 5% CO_2_. Human lung bronchial epithelial cells 16HBE14o- (named CTRL cells), originally developed by Dieter C. Gruenert, were provided by Luis J. Galietta, (Telethon Institute of Genetics and Medicine—TIGEM, Napoli) and cultured, as recommended, in Minimum Essential Medium (MEM) Earle’s salt, supplemented with 10% FBS and 1% penicillin/streptomycin at 37 °C and 5% CO_2_. Myriocin (Myr) treatments were performed at a concentration of 50 µM for the indicated time lengths in 100 mm dishes plated at 1 × 10^5^ cells/each. At least triplicate samples for each experiment were performed.

### 2.2. Aspergillus Fumigatus Culture

The reference strain *A. fumigatus* Af293 (ATCC MYA-4609, CBS 101355) was used in the study. Frozen conidia were streaked out from glycerol stocks stored at −80 °C on fresh Potato Dextrose Agar (PDA) slant and incubated at 30 °C for 72–96 h, until sporulation. Conidia were then harvested, suspended in 0.01% Tween-20, washed in phosphate-buffered saline (PBS), and counted by hemocytometer.

### 2.3. Cells Infection

We plated 1 × 10^6^ cells, either CF or CTRL, in 100 mm Petri dishes and grew them overnight in 5 mL of appropriate medium. The next day, cells were pre-treated with 50 μM Myr for 1 h and then infected with *A. fumigatus* conidia with a multiplicity of infection (MOI) of 1:100 for 1 h. Cells were washed three times in PBS to eliminate non-engulfed conidia and incubated for 3 h with 5 mL medium containing or not 50 μM Myr. Cells were collected at this time, or after 12 h, for subsequent analysis.

For the killing assay, patients’ monocytes were pretreated with Myr (50 µM) or medium alone for 1 h at 37 °C and then infected with *A. fumigatus* conidia (MOI 1:1). One hour after infection, cells were washed twice with PBS to remove non-internalized conidia, and incubated for a further 4 h to allow conidia killing. Cells were then harvested, counted, and lysed by osmotic shock to recover live internalized conidia. Cellular lysis was confirmed by microscopy. The supernatant was mixed vigorously, serially diluted in PBS, and plated onto Sabouraud Dextrose Agar (SAB). Plates were incubated at 37 °C, and colony forming units (CFU) were counted after 48 and 72 h of growth.

### 2.4. PBMC Isolation and Infection

We collected 43 blood samples from CF patients referred to the Ancona Cystic Fibrosis Centre, 22 homozygotes for the F508del mutation of the *CFTR* gene and 21 compound heterozygotes with one F508del allele (as shown in Appendix A) (United Hospital Le Torrette, Ancona, Italy, CE Regionale Marche -CERM-, protocol number 2016 0606OR).

Peripheral blood mononuclear cells (PBMCs) were freshly isolated using Leucosep protocol and stored at −80 °C until use. To avoid the influence of anti-CD14-coated microbeads selection on cytokine production, we isolated monocytes by plastic adherence [27]. Briefly, 1 × 10^6^ PBMCs per well were seeded into 6-well plates (Corning Inc. Costar, New York, NY, USA) and allowed to adhere in a 5% CO_2_ incubator at 37 °C for 2 h in 2 mL Roswell Park Memorial Institute (RPMI) 1640 medium supplemented with 10% FBS and 1% penicillin/streptomycin. Free-floating cells were then removed by washing and the adhering monocytes were used for *A. fumigatus* infection as described above for cell lines.

### 2.5. qRT-PCR

One microgram of purified RNA was reverse transcribed to cDNA. The amplification was performed for the following target genes: *IL-1B*, *CXCL8*, *IL-10*, *NOD2*, *TLR2*, *TLR7*, *TBK1*, *OPTN*, *TFEB*, *LAMP2a*, *TP53INP1, TMEM59, HIGD1A*. Relative mRNA expression of target genes was normalized to the endogenous *GAPDH* control gene and represented as fold change versus control, calculated by the comparative CT method (∆∆CT Method). The analysis was performed by referring to control values that did not significantly differentiate (triplicate samples, and their standard deviation value divided by their mean value was <1). Appendix A shows the ∆∆CT analysis of the target genes in basal condition, with CF cells versus CTRL cells. Real-time PCR was performed by SYBR Premix Ex Taq^™^ II (Takara); primer sequences are available on request.

### 2.6. Statistical Analysis

Data are expressed as mean ± SE, calculated from experimental replicates. Data significance was evaluated by two-way ANOVA followed by Bonferroni correction (*p* < 0.05) or Wilcoxon *t*-test (for paired samples), as specified in the figure legends. Statistical analysis was performed by GraphPad InStat software (La Jolla, CA, USA) and graph illustrations were generated by GraphPad Prism software (La Jolla, CA, USA).

### 2.7. RNA Extraction and Sequencing

Total RNA was isolated from harvested cells and patients’ samples with the ReliaPrep™ Miniprep RNA extraction system (Promega), according to the manufacturer instructions.

Sequencing was performed on Illumina NextSeq using the SMART-Seq protocol for the preparation of the libraries, obtaining an average of 15 million single-end reads per sample. All sequences were of fixed length 75 bp, and of a high quality for a single base.

Twenty four samples were analyzed: 12 derived from a cell line of CF pulmonary epithelium, and 12 from a cell line of healthy pulmonary epithelium (CTRL). For each group, the following conditions were analyzed: (1) basal condition (CTRL or CF); (2) infection with *A. fumigatus* (Asp); (3) myriocin treatment (Myr); (4) infection and treatment (Asp-Myr). Three biological replicates were used for each condition.

RNA was also extracted from a selection of *A. fumigatus*-infected monocytes from 11 homozygous and 9 heterozygous patients. For each subject, we analyzed 2 conditions: (1) samples infected with *A. fumigatus* (untreated); (2) samples infected with *A. fumigatus* and treated with Myr (treated).

### 2.8. Exploratory Data Analysis

In order to evaluate samples variability, we applied principal component analysis (PCA) as a tool to make assumptions. In particular, we performed PCA starting from regularized log-transformed counts matrices created by the rlog function implemented in the DESeq2 package [28].

### 2.9. RNA-Seq Data Analysis

Raw single-end reads were aligned to the human reference genome (GRCh38) using STAR [29], and only uniquely mapping reads were considered for downstream analyses. Reads were assigned to genes with featureCounts [30], using the Gencode primary assembly v.31 gene transfer file (GTF) as reference annotation for the genomic features. Raw counts matrices were then processed with the R/Bioconductor differential gene expression analysis packages Deseq2 and EdgeR [31] following standard workflows. In particular, for the monocyte dataset we set up a paired analysis, modeling gene counts using the following design formula: ~patient + condition. For cell lines, due to the high variability between and within cell lines, we split the dataset into specific subsets. In particular, in both CTRL and CF cells, we evaluated the effect of *A. fumigatus* (Asp) infection and of Myr with and without infection. Genes with adjusted *p*-values less than 0.01 (cell lines) and 0.05 (monocytes dataset) were considered differentially expressed (DEGs). Downstream analyses, including gene set enrichment analysis (GSEA) and over-representation analysis (ORA), were performed with the ClusterProfiler R/Bioconductor package [32] using a list of databases including Gene Ontology (GO), KEGG, Reactome pathway, and the Molecular Signatures Database (MsigDB). Enriched terms with an adjusted *p*-value < 0.05 were considered statistically significant. Charts and images were produced using the ggplot2 R package.

We performed protein–protein interaction (PPI) networks functional enrichment analysis using string-db [33] web site. In particular, starting from the default setting we applied the following changes: removal of text-mining flag from active interaction sources, selection of highest confidence interaction scores, and hiding of disconnected nodes from graphical output.

## 3. Results

### 3.1. Myriocin Transcriptionally Regulates and Ameliorates the Response to A. fumigatus Infection in CF-Patients-Derived Monocytes

In order to study the effects of Myr on the immune response against fungal infection in CF, monocytes derived from the PBMCs of CF patients bearing homozygous or heterozygous ΔF508 CFTR mutation were infected in vitro with *A. fumigatus* and either treated or not treated with Myr. Myr treatment significantly increased *A. fumigatus* killing compared with untreated infected CF monocytes (Figure 1).

Exploratory data analysis showed a patient-specific bias that we took into account by using a paired differential expression model, as described in the Methods section. After correcting for batch effect (patient), we clearly observed a separation on the second principal component (Figure 2A). Differential gene expression analysis identified 1460 DEGs between Myr-treated and untreated cells, in particular 563 upregulated and 896 downregulated genes in Myr-treated cells (Figure 2B).

Starting from the DEG list, we performed over-representation analysis using different gene sets, including Gene Ontology (GO), Kyoto Encyclopedia of Genes and Genomes (KEGG), and Molecular Signatures Database (MSigDB) terms, in order to identify the presence of enriched functional categories. In particular, we observed categories related to CF disease as inflammation, autophagy, lipid metabolism, and infection (Figure 3A). We then analyzed the proportion of differentially upregulated and downregulated genes and highlighted the most significantly modulated ones. We noticed a significantly higher number of upregulated genes under Myr treatment (Figure 3B).

Moreover, we performed protein–protein interaction (PPI) networks and functional enrichment analysis (STRING) of both up- and downregulated DEGs. Interestingly, we noticed that PPI network analysis of upregulated genes highlighted the presence of two main clusters of interacting proteins enriched in histone molecules and heat-shock proteins, respectively (Figure 3C).

By studying the most significant DEGs, we observed that genes belonging to the above-mentioned clusters are related to the ontologies indicated in Figure 3A and to the two main clusters of interacting proteins (Figure 3C). Moreover, their activity is relevant and novel in CF infection. A number of genes found to be upregulated belong to the histone cluster family and their corresponding proteins, which can locate within the cytosol and are recognized for exerting antimicrobial activity [34,35,36,37,38,39,40] (Table 1).

To note that Myr also induced the upregulation of several genes encoding for heat-shock proteins: Hsp70, Hsp90, Hsp40 (DnaJ), and Hsp-interacting proteins (Table 2), as already highlighted by PPI network analysis (Figure 3C).

Such proteins are directly involved in chaperone-mediated autophagy, which sustains the autophagic activity. In addition, we identified an increased transcription of the *PNPLA3* gene that codes for a lipase supporting lipid storage degradation in autophagolysosomal compartment [41].

Moreover, Myr inducedthe upregulation of LC3IIB (*MAP1LC3B2*), which belongs to the LC3B family, required for cargo and autophagy protein recruitment and for autophagosome nucleation [42].

### 3.2. CF and Control Airways Epithelial Cell Lines Exhibit a Different Transcriptional Response to Infection

Chronic inflammation contributes to defective pathogen clearance in CF airways, and epithelial cells offer the first barrier to microbial infection by exerting pathogens’ uptake and killing. In order to better understand the therapeutic effect of sphingolipid synthesis inhibition in CF, we studied the transcriptomic profile of CF bronchial epithelial cells in the early phase of infection (4 h). Hence, we performed an RNA-sequencing analysis of CF and CTRL bronchial epithelial cells after infection with *A. fumigatus.*

As shown in Figure 4A, infected CF cells exhibited a stronger transcriptional response to fungal infection compared to CTRL cells, considering both fold changes and differentially expressed genes (DEGs). In response to infection, we identified 3954 DEGs in CTRL cells and 5109 DEGs in CF cells (Figure 4B).

### 3.3. Myriocin Treatment Modulates CF Expression Profile under Infection

Next, we evaluated the effect of myriocin on gene expression in CF and CTRL cells, both in basal and in *A. fumigatus* infection conditions. We observed a significantly higher number of DEGs in Myr-treated CF (MyrCF, 1629 DEGs) than in CTRL cells (MyrCTRL 55 DEGs) (Figure 5A). Myr treatment in *A. fumigatus*-infected CF cells (MyrAspCF/Asp) significantly modulated only 62 DEGs versus 240 DEGs in infected CTRL cells (MyrAspCTRL/Asp) (Figure 5B). This could be related to an attenuated transcriptional effect of the compound due to the massive transcriptional change induced by *A. fumigatus* infection in CF cells (AspCF), which was significantly stronger than in CTRL cells (AspCTRL) (see above, Figure 4).

To get insight from the obtained data, we created Venn diagrams showing the common DEGs, deriving from the three different comparisons: (i) infected CTRL cells versus uninfected (AspCTRL); (ii) infected CF cells versus uninfected (AspCF); (iii) infected and Myr-treated CF cells versus uninfected and untreated (MyrAspCF).

We found 292 genes that were differentially expressed in both CTRL and CF cells in response to fungal infection (AspCTRL and AspCF). When CF cells were treated with Myr (MyrAspCF), the number of common regulated genes increased up to 439, indicating that Myr drives CF response to infection and partially restores the CTRL expression profile (Figure 5C). Finally, we compared the genes that found to be regulated by Myr in infected airways epithelial CF cells (MyrAspCF) and in infected CF patients’ derived monocytes (treated-infected versus untreated-infected) (Figure 5D). Regardless of the different origins of the cells, we found a common upregulation of HSP90AA1 and ZFAS1, two genes belonging to heat shock and zinc fingers proteins families, involved in chaperone-mediated autophagy (previously described, Table 2, Section 3.1, [40,43,44]). In addition, the expression of another 11 genes was commonly modulated, although at minor extent in respect to the above discussed.

### 3.4. Myriocin Activates Gene Sets Involved in Inflammation, Infection, Autophagy/Proteostasis, and Lipid Metabolism in CF Bronchial Epithelial Infected Cells

In order to evaluate the effect of Myr on infected CF cells, we performed GSEA using logFC pre-ranked list of MyrAspCF/Asp DEGs, thus evaluating only Myr-related transcriptional activities and excluding infection-induced transcriptional modification. Results indicate that the Myr treatment modulates the expression of genes involved in autophagy/proteostasis, lipid metabolism, and in response to infection and inflammation. In particular, as highlighted in Figure 6, Myr treatment upregulated the genes involved in autophagy/proteostasis and lipid metabolism, whereas it downregulated genes related to inflammation and infection processes.

Considering a significant adjusted *p*-value threshold of 0.05, we observed 139 DEGs in infected CF cells in response to treatment. Among such DEGs we identified specific genes related to the molecular processes that are regulated by Myr, as evidenced by GSEA. Other than increasing IL1β, a primary cytokine in CF disease and its defective response to infection [45,46], Myr upregulated the expression of the transmembrane protein TMEM59, which induces the LC3 labelling of endosomal vesicles, stimulating their fusion with lysosomes, in response to autophagy and xenophagy [47]. Moreover, Myr upregulated the p53-inducible protein 1 (*TP53INP1*) gene, which also promotes autophagy by interacting with autophagy-related protein family (ATG) [48]. The expression of SNX14, involved in the regulation of autophagy and lipid metabolism, was significantly increased by Myr [49,50], as well as that of hypoxia inducible gene 1 (*HIGD1A*), involved in oxidative stress and lipotoxicity protection [51]. At the same time, Myr downregulated ACACA, the rate-limiting enzyme regulating de novo fatty acid synthesis, whose increased activity has been associated with inflammation and CFTR deficiency [52,53]. We next validated the upregulation of three of the above reported genes by RT-PCR and demonstrated the increase of their expression in infected and Myr-treated CF cells (Figure 7).

In view of our previous results on Myr transcriptional effects in CF bronchial epithelial cells [5,8,25], we evaluated a delayed expression of specific marker genes involved in the inflammatory process and in the response against microbial infection in CF cells infected with *A. fumigatus*, treated or untreated with Myr. By real-time PCR (RT-PCR), we proved that the expression of pro-inflammatory interleukin-1β (*IL-1β*) and *IL-8* chemokines, known to be upregulated in CF [25], were significantly reduced by Myr, in both basal conditions and in *A. fumigatus*-infected CF cells, whereas the expression of anti-inflammatory *IL-10* was upregulated by the compound (Figure 8A–C).

Pathogen recognition receptors (PRRs) are responsible for the identification of antigens, and can mediate their lysosomal clearance. We observed that Myr treatment enhanced the expression of the *NOD2, TLR2*, and *TLR7* in infected CF cells (Figure 8D–F). Myr reduction of sphingolipid synthesis regulates lipid metabolism by enhancing fatty acids oxidation and reducing the overall cell amount of glycerolipids and cholesterols [5]. Myr’s action on lipid-energy homeostasis is sensed as a stress that drives TFEB activation and transcriptional activities that sustain lipid consumption and autophagy induction [5]. We observed an increased expression of *TFEB* in infected CF bronchial epithelial cells treated with Myr compared to untreated infected cells (Figure 8G). TBK1 phosphorylation and interaction with optineurin (OPTN) promote autophagy-mediated pathogen clearance, namely xenophagy [54,55,56]. We observed that *A. fumigatus* infection reduced the expression of *TBK1* while Myr treatment significantly rescued it (Figure 8H). Myr was also able to increase *OPTN* expression in infected CF cells (Figure 8I). Finally, Myr induced a significant increase of *Lamp2a* expression, which was reduced in *A. fumigatus*-infected CF cells, confirming the TFEB and the autophagy-related increase in lysosome formation (Figure 8L).

## 4. Discussion

Our study demonstrates that the defective response to infection in CF is related to a dysfunction in autophagy, inflammation resolution, and lipid metabolism, which are known to be caused by mutated CFTR [1,2,3,4,5]. Dyslipidemia has been associated with CF disease, and it is characterized by a reduced absorption and increased synthesis of lipids [1,2,53,57,58,59,60,61,62]. Moreover, cholesterol and the inflammatory lipid ceramide have been shown to accumulate in CF peripheral organs, in particular at the airways level [4,23,26,63,64,65,66]. Although altered lipid metabolism is a common feature of chronic inflammatory diseases, the contribution of lipids in CF pathophysiology is still to be fully elucidated. We previously demonstrated that Myr—a specific inhibitor of sphingolipid de novo synthesis—reduces the accumulation of not only ceramide but also most lipid species in CF cells [5]. By impairing sphingolipid synthesis, Myr induces the activation of a stress response that initiates with the TFEB-induced transcriptional program, sustained by PPARs and FOXOs transcription factors, aimed at increasing lipid oxidation and promoting autophagy [5]. Its modulatory action results in an overall decline of CF hyperinflammation, as a consequence of a reduced expression of pro-inflammatory cytokines, and in an ameliorated defensive response to infection, driven by the increase in xenophagy-activating PRR expression [8,25], known to be downregulated in CF [67].

Monocytes play a crucial role in pathogen eradication and CF prominent susceptibility to recurrent infections, largely relying on altered monocytes response. Therefore, we studied peripheral-blood-derived monocytes from CF patients, bearing either homozygous or compound heterozygous ΔF508 mutation of CFTR, by in vitro infection with *A. fumigatus*. Myr treatment shaped a significantly different transcriptional process in infected monocytes, modulating the expression of genes involved in inflammation, response to infection, lipid metabolism, and autophagy. Among differentially expressed genes, we identified two functional clusters in terms of significant response to Myr that have not previously been associated to CF (Table 1 and Table 2). Myr upregulated the expression of several proteins belonging to the histone family. Other than fundamental components of eukaryotic chromatin, histones and histone fragments display antimicrobial activities [34], either by being secreted and reacting against extracellular pathogens or by accumulating in the cytosol and binding intracellular infectious agents [38]. Cytosolic histones elude proteolysis thanks to the binding to lipid cytosolic storage [35], are released upon interaction with pathogens, and are processed to act as antimicrobial peptides against bacteria and fungi [36,37,39]. The observation that infection increases this cytosolic histone-related fraction [35] supports our hypothesis that, possibly by modulating lipid metabolism, Myr promotes histones transcription in infected cells in order to boost their cytosolic pool, which is endowed with antimicrobial activity. Moreover, Myr upregulated the expression of a number of heat-shock proteins (HSP) belonging to the 70 family (Hsp70), 90 family (Hsp90), and DnaJ family (Hsp40). HSPs take part in the response to infection by receptor-mediated activation of the innate immune response and by participation in the antigen presentation for the adaptive immune response [68,69], which is also defective in CF [6]. Hsp70 and Hsp90 play an important role in chaperone-mediated autophagy (CMA) and pathogen recognition [44,70,71]. Proteins degraded by CMA are identified in the cytosol by a chaperone complex which includes Hsp70 [72]. Upon binding to the target, Hsp70 [73] and Hsp90 [44] interact with the lysosome-associated membrane protein type 2A (LAMP-2A), which we observed to be upregulated by Myr. DNAJ proteins (Hsp40) regulate Hsp70 chaperones by stimulating ATP hydrolysis [74]. Thus, Myr’s action on HSPs is aimed at sustaining innate and adaptive immunity as well as autophagy-related unfolded proteins and pathogens clearance in infected CF monocytes. Moreover, Myr increased the expression of the *PNPLA3* gene, which encodes for a lipase responsible for the mobilization of intracellular fat storage [75] and degradation in the autophagolysosomes. This process, named lipophagy, is directly promoted by TFEB activation [41]. Hence, Myr might sustain both classic autophagy, as we previously demonstrated [5], and chaperone-mediated autophagy, driving not only the autophagic pathway, but also TFEB-induced lysosome biogenesis, as revealed by increased *Lamp2a* expression. In agreement with previous data suggesting that Myr enhances killing ability in an *A. fumigatus*-infected CF bronchial epithelial cell line [8], we observed a significant increase of conidia killing in *A. fumigatus*-infected CF monocytes. This latter observation indicates that reducing inflammatory lipid accumulation and promoting autophagy is an effective therapeutic approach to correcting proteinopathy stress, mainly related to ΔF508 mutation of CFTR, and to restoring an effective response against infection.

Next, we extended our previously published data by investigating the effect of sphingolipid synthesis inhibition in the first phase of microbial infection in the airways. In order to better understand whether Myr could modulate CF response to infection, we analyzed the whole-transcriptome modification induced by *A. fumigatus* in a CF bronchial epithelial cell line and a healthy control counterpart, either treated or untreated with Myr. Our data provide clear evidence that a profound modification in gene expression, triggered by infection, differentiates CF from healthy cells, suggesting that the disease itself causes higher sensitivity to infection. Indeed, healthy cells responded to the same stimulus by engaging a smaller number of genes. This evidence is in line with CF patients’ defective ability to elicit a coordinated defensive response against infection. Indeed, we observed a stronger modulation of CF cells’ transcriptional activity by Myr treatment, whereas the impact on control cells was milder. These data are in agreement with our previous observation on the effects of Myr treatment on inflamed cells or organs with respect to their healthy counterparts [5,25,76]. We suggest that sphingolipid de novo synthesis is enhanced under stress conditions such as proteinopathy and inflammation. In a homeostatic state, Myr inhibition of the sphingolipid synthesis rate-limiting enzyme serine palmitoyl transferase (SPT) may be buffered by a consequent spontaneous modulation of Nogo and ORMs, two enzymes that normally control SPT activity [77,78]. When stress or inflammation upregulate the sphingolipid synthesis pathway, the effect of Myr is more pronounced and is perceived as an alarm signal that drives defense response by moving the metabolic resources via lipid mobilization and autophagy. During an infectious process, CF cells’ transcriptional activities are perturbed to a higher extent than control cells. Nonetheless, Myr treatment increased the number of genes that were commonly regulated by infection in CF and healthy cells, thus driving the CF phenotype closer to that of control cells. This confirms the hypothesis that inhibition of sphingolipid synthesis activates a profound transcriptional modification aimed at inducing stress tolerance and resilience as well as pathogen resistance in CF. This concept is crucial in any chronic inflammatory disease, in which the possibility of counteracting the cellular stress is the required therapeutic effect. Our data indicate that the Myr-induced effects on infected CF cells involve genes mediating inflammation, immune response, autophagy, and lipid metabolism. Among genes whose expression was significantly affected by Myr in CF cells during infection, we identified a few that are related to the significantly regulated pathways and that may indicate novel targets for CF therapy. A specific form of LC3-associated phagocytosis can be activated by TLR signaling during the phagocytosis of fungal and bacterial pathogens: LC3 binds to the cytosolic side of the TLR-induced endosome, which evolves into autophagosomes [68]. Mediating this mechanism, TMEM59 and TMEM166 interact with ATG16L1, become incorporated into phagosomes’ membranes, driving the activation and lipidation of LC3, and thus the maturation into autophagosomes [69]. Similarly, SNX14 proteins generally localize to endosomes’ membranes via the PX domain, which binds to phosphoinositides (PtdIns), and this is reported to promote autophagy [49,50]. The expression of both SNX14 and TMEM59 was enhanced by Myr upon infection, suggesting that the compound drives a pathogen-clearance-related autophagy. Moreover, altered SNX14 function has been associated with reduced cholesterol esters, suggesting an alteration in neutral lipid metabolism and possibly reduced lipid delivery to droplets storage [50]. Accordingly, Myr reduced the expression of the gene encoding for acetyl-CoA carboxylase (ACACA), the rate-limiting enzyme involved in fatty acids synthesis and regulated in opposition to their oxidation. ACACA has been associated with inflammation [52] and most notably, its activity is enhanced in CFT-deficient cells [53], in line with the increased sphingolipid synthesis in CF chronic inflammation [5,25]. Myr treatment also increased the expression of hypoxia-inducible gene domain family member 1A (*HIGD1a*). Its gene product was recently reported to protect cells from hypoxia and from lipotoxicity related to fat oxidation impairment. Indeed, HIGD1a decreases oxygen radical production, helping to maintain a normal mitochondrial function [51]. A reduced autophagy paired with lipid accumulation because of decreased oxidation rate was already observed in CF cells and rescued by Myr [5]. Thus, Myr could possibly enhance *SNX14* to induce autophagy and lipid consumption, and it may upregulate *HIGD1a* to reduce the oxidative stress caused by proteinopathy and lipid accrual in infected CF cells. Therefore, Myr is able to directly target specific genes whose functions have not been previously associated to CF, but that sustain its pathophysiology.

In addition, lipid synthesis inhibition is a strategy to fight microbial infections that are known to take advantage of cell lipid storage, and ACACA inhibitors have been designed to develop antimicrobial strategies in infectious diseases [79].

Finally, we investigated the possibility of Myr-induced transcriptional activation of TFEB and of downstream target genes in *A. fumigatus*-infected CF bronchial epithelial cells. As expected, TFEB transcription was increased by Myr, suggesting that lipid catabolism and autophagy are enhanced even under infection conditions. A growing body of evidence highlighted that TBK1-OPTN signaling is pivotal for the initiation and resolution of the innate immune responses, and in eliciting autophagy [55,56,80]. Myr significantly increased the expression of these two proteins, both in uninfected and infected cells, thus promoting internalized pathogen clearance. To note, Myr upregulated the expression of *LAMP2A* (lysosomal-associated membrane protein 2A), encoding for a membrane glycoprotein involved in autophagy, which is directly promoted by TFEB induction of lysosomes biogenesis.

## 5. Strength and Limitations of the Study

The strength of the present study is that it presents data deriving from a significant number of CF patients, variable for sex and age, affected by ΔF508 mutation of the *CFTR* gene (homozygous and compound heterozygous). The limitations include the fact that part of the study was conducted on epithelial cell lines, and further studies on primary epithelial cells are needed to corroborate the obtained data. Moreover, we cannot exclude that patient PBMC storage and in vitro culturing could impact the cell transcriptional response. Next, although the used *Aspergillus fumigatus* strain (Af293) is a clinical strain derived from invasive pulmonary aspergillosis, a genome-sequenced biofilm producer, the in vitro infection procedure may not completely resemble in vivo infection, and the host response may differ in terms of intensity and timing of gene expression modulation. Similarly, we cannot predict whether Myr administration would be efficient in modulating chronic infections in patients. From a technical point of view, gene expression was carried out by normalizing data with a unique housekeeping gene (*GAPDH*), whereas the use of three different housekeeping genes would ensure the highest reliability of transcriptional evaluation.

## 6. Conclusions

CF bronchial epithelial cells display a profound difference in the transcriptional profile compared with their normal counterpart, and infection sharpens this diversity. We demonstrated that by modulating the biosynthesis of sphingolipids by Myr, a metabolism drift occurs which is aimed at fueling energy to act against stress, and includes autophagy and lipid consumption. This is driven by a transcriptional program that significantly modifies cellular phenotype and amends part of the differences between CF and control, therefore ameliorating the response against infection and pathogens clearance in both a CF cell line [8] and monocytes. We speculate that lipid metabolism is deeply altered in CF patients, possibly due to the chronic and systemic inflammation associated with the disease. Lipids accumulation, or simply deregulation in their storage/consumption, may contribute to the defective immunity of CF patients. Literature evidence reports dyslipidemia in CF patients, measured both in blood and peripheral organs [2,81,82]. We conclude that more attention should be devoted to the alteration of lipid metabolism in CF.

## Figures and Tables

**Figure 1 cells-09-01845-f001:**
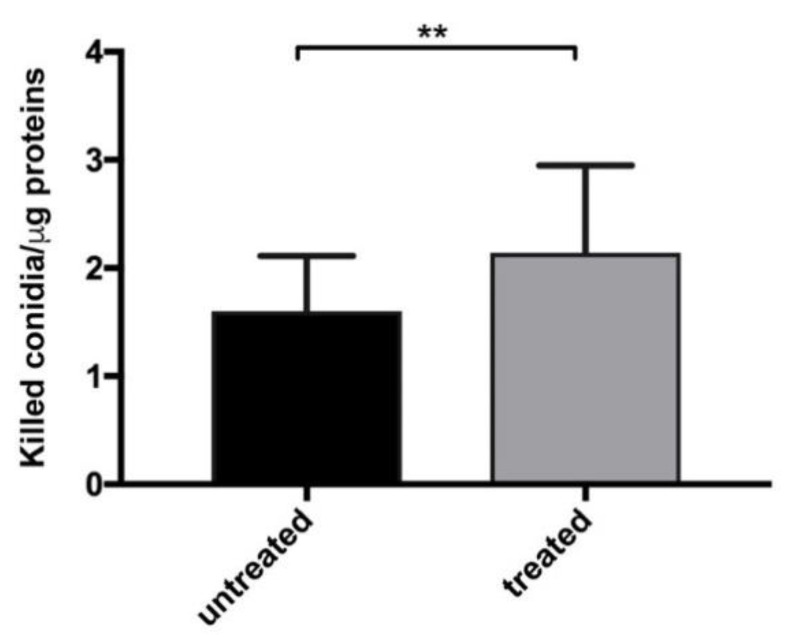
Myriocin (Myr) treatment improved monocyte killing ability. *Aspergillus fumigatus* conidia were added to patients’ monocytes, untreated or treated with Myr, and cells incubated for 1 h to allow conidia internalization. Non-internalized conidia were then washed, and monocytes were incubated for a further 4 h before undergoing lysis. Live conidia were counted by plating cell lysate on solid medium. To normalize the results, data are expressed as killed conidia/µg protein. Myr partially rescued cystic fibrosis (CF) monocytes’ killing ability (** *p* < 0.01 Wilcoxon *t*-test).

**Figure 2 cells-09-01845-f002:**
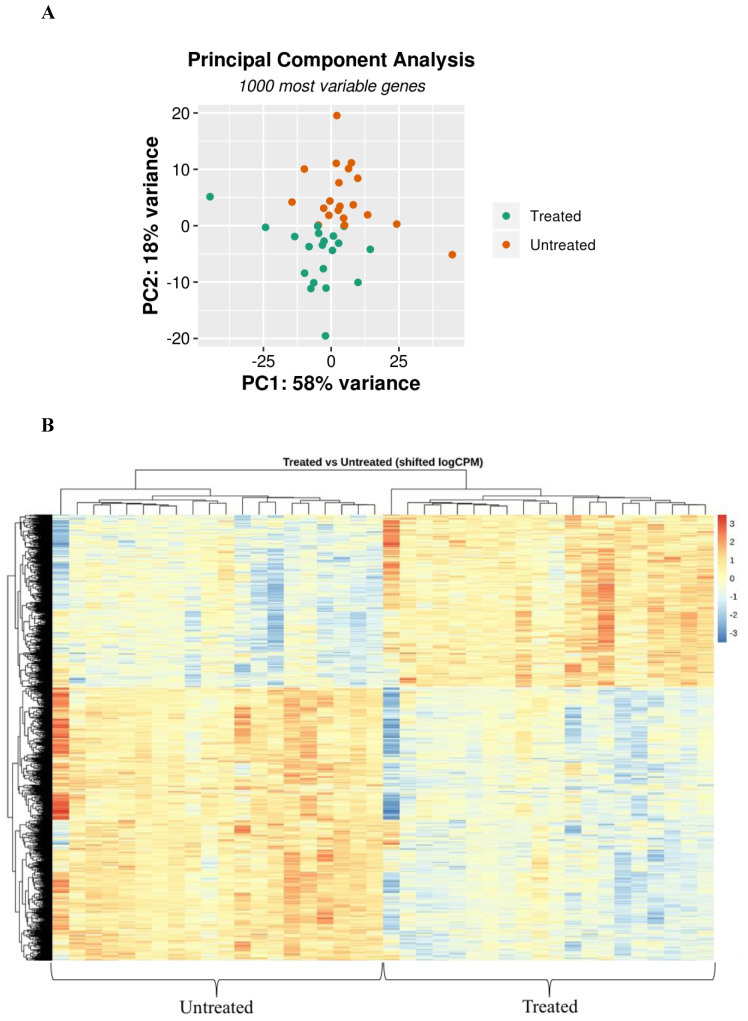
PCA and differential gene expression (DGE) induced by myriocin treatment in CF-patients-derived monocytes. (**A**) PCA was calculated considering the 1000 most variable genes across the dataset. Log-normalized counts were “shifted” for patient in order to take into account the experimental design adopted for DGE analysis. (**B**) Heatmap of DEGs with adjusted *p*-value < 0.05 resulted from the comparison between treated and untreated samples. The data used for creating this image were logCPM-corrected for patient batch and row-scaled (heatmap package).

**Figure 3 cells-09-01845-f003:**
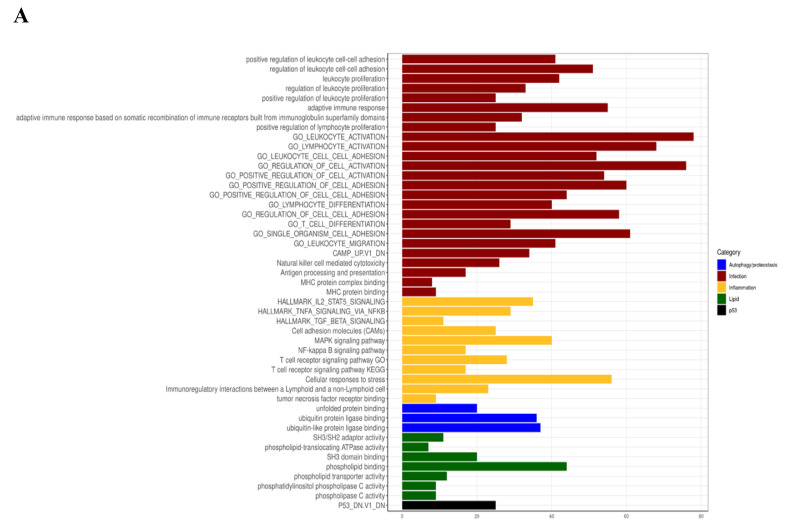
(**A**) CF-related over-represented terms. DEGs obtained from Myr-treated and untreated *A. fumigatus*-infected CF monocytes; the comparisons were used for performing over-representation analysis (ORA) using GO, KEGG, and MSigDB terms. Here are shown the enriched terms (adjusted *p*-value < 0.05) matching the top functional categories related to CF including autophagy, infection, inflammation, and lipid metabolism. On the *x*-axis, the number of DEGs belonging to the enriched term. (**B**) Volcano plot showing DEGs distribution between up- and downregulated genes with labels for the top 20 significant DEGs. On the *x*-axis, the log2Fold change, and on the *y*-axis the adjusted *p*-value (−log10). (**C**) STRING analysis of up regulated DEGs in treated versus untreated comparison.

**Figure 4 cells-09-01845-f004:**
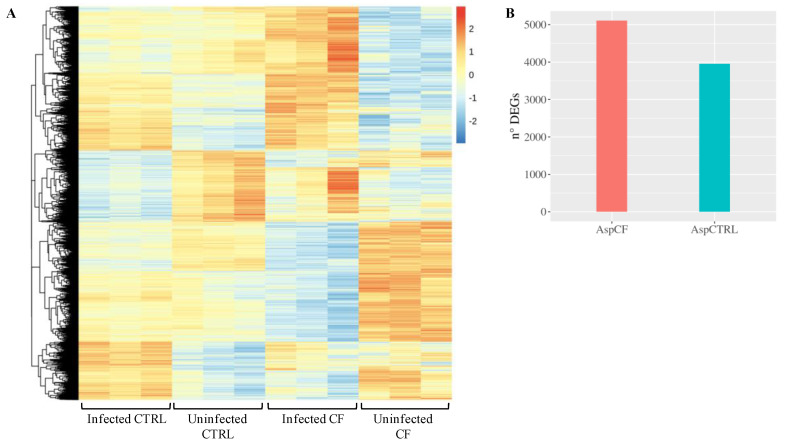
Transcriptional response to infection. (**A**) The heatmap represents the expression (rlog row-scaled) of DEGs obtained in the AspCTRL and AspCF comparisons. (**B**) DEGs numbers. AspCTRL: DEGs between infected and uninfected CTRL cells; AspCF: DEGs between infected and uninfected CF cells.

**Figure 5 cells-09-01845-f005:**
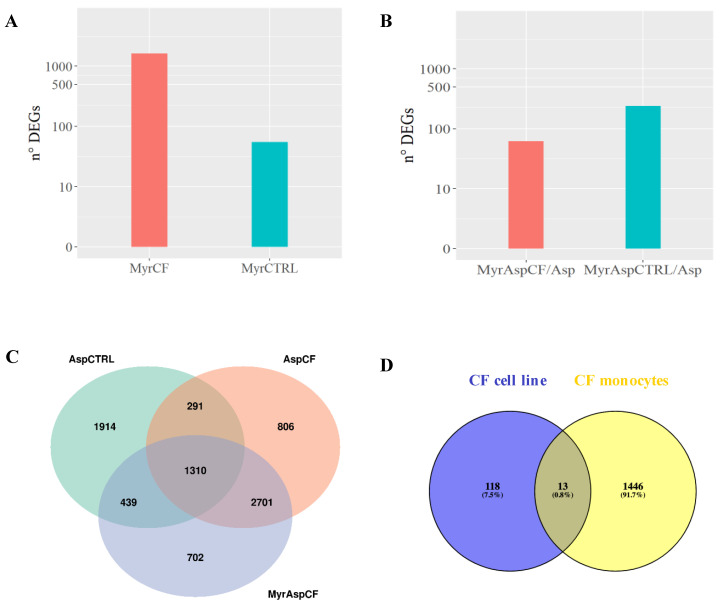
Effect of myriocin on CF and CTRL cells in infected and infection-free environments. (**A**) MyrCF indicates DEGs between Myr-treated CF cells and untreated CF cells; MyrCTRL indicates DEGs between treated CTRL cells and untreated CTRL cells; (**B**) MyrAspCF/Asp indicates DEGs between Myr-treated infected CF cells and untreated infected CF cells; MyrAspCTRL/Asp indicates DEGs between Myr-treated infected CTRL cells and untreated infected CTRL cells. (**C**) Venn diagram of CRTL and CF cells in response to infection (AspCTRL and AspCF) and infected CF cells in response to treatment (MyrAspCF). MyrAspCF: DEGs between Myr-treated infected cells and uninfected untreated CF cells. (**D**) Venn diagram of CF cells and CF monocytes: DEGs between Myr-treated infected CF cells and Myr-treated infected monocytes from CF patients.

**Figure 6 cells-09-01845-f006:**
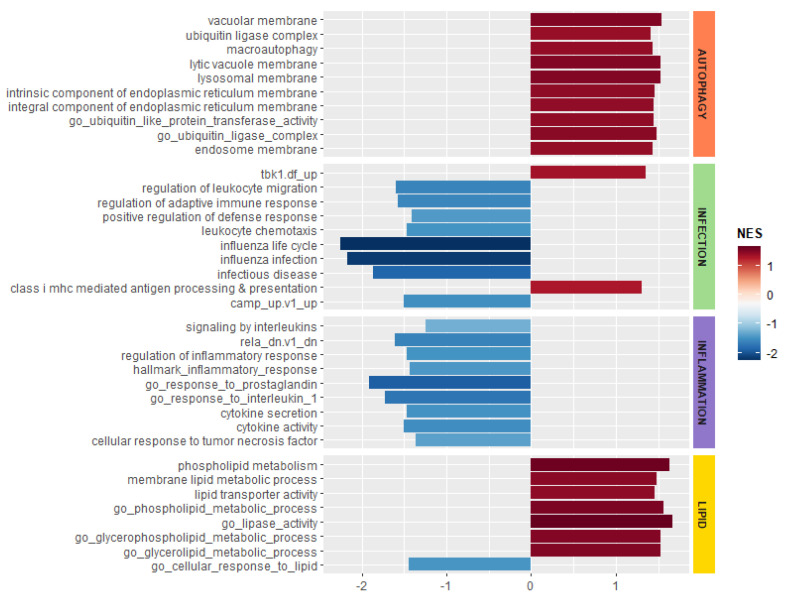
Enriched gene set enrichment analysis (GSEA) terms—functional categories. This chart shows the results of GSEA analysis on functional categories related to CF pathology using pre-ranked logFC list genes from MyrAsp/Asp CF analysis. In red, the terms with positive normalized enriched score (NES), upregulated by myriocin, whereas in blue the terms with negative NES, downregulated by myriocin. GSEA terms include GO, MutSigDB, and KEGG pathway terms.

**Figure 7 cells-09-01845-f007:**
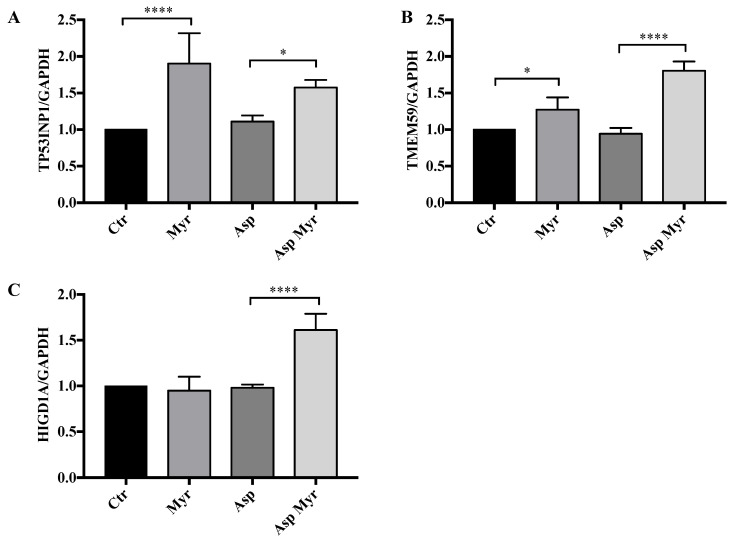
RNA-seq validation by RT-PCR quantification of the expression of genes involved in response to infection in CF cells: (**A**) transmembrane protein TMEM59; (**B**) p53-inducible protein 1, *TP53INP1*; (**C**) hypoxia inducible gene 1, *HIGD1A.*
*GAPDH* was used as a housekeeping gene. Data, derived for triplicate samples, are expressed as mean ± SE (* *p* < 0.05; **** *p* < 0.0001); two-way ANOVA followed by Bonferroni correction was used for all data.

**Figure 8 cells-09-01845-f008:**
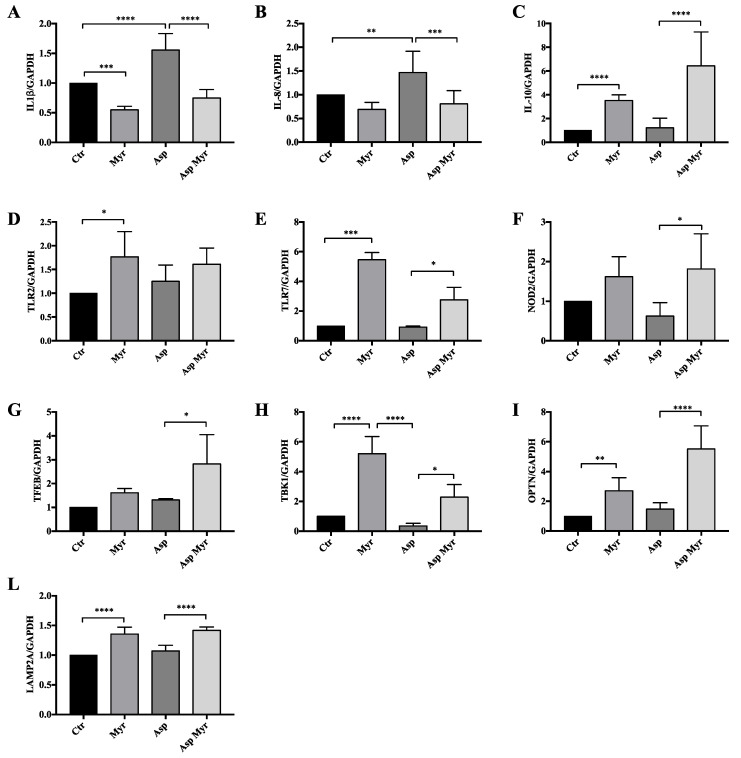
Quantification of the expression of genes involved in inflammation, response to infection, and autophagy: (**A**) pro-inflammatory *IL1β* interleukin; (**B**) pro-inflammatory *IL-8* chemokine; (**C**) anti-inflammatory *IL10* interleukin; (**D**–**F**) pathogen recognition receptors (PRRs): *NOD2*, *TLR2*, and *TLR7*; (**G**) *TFEB*; (**H**) *TBK1*; (I) *OPTN*; (**L**) *LAMP2a* by qRT-PCR in *A. fumigatus*-infected and uninfected CF cells, treated and untreated with Myr (12 h after infection). *GAPDH* was used as a housekeeping gene. Data, derived for triplicate samples, are expressed as mean ± SE (* *p* < 0.05; ** *p* < 0.01; *** *p* < 0.001; **** *p* < 0.0001); two-way ANOVA followed by Bonferroni correction was used for all data.

**Table 1 cells-09-01845-t001:** Cluster of histone protein DEGs between Myr-treated and untreated CF-patients-derived monocytes.

Symbol	Log2FoldChange	Padj	Description	Ensembl
HIST1H2BH	1.86	8.07 × 10^−5^	histone cluster 1 H2B family member h	ENSG00000275713
HIST1H2BG	1.56	0.001394033	histone cluster 1 H2B family member g	ENSG00000273802
HIST1H2BO	1.54	0.021985759	histone cluster 1 H2B family member o	ENSG00000274641
HIST1H2BE	1.53	0.007600047	histone cluster 1 H2B family member e	ENSG00000274290
HIST1H2BJ	1.29	0.002758228	histone cluster 1 H2B family member j	ENSG00000124635
HIST1H2BD	1.21	0.001620018	histone cluster 1 H2B family member d	ENSG00000158373
HIST1H4C	1.14	0.004057245	histone cluster 1 H4 family member c	ENSG00000197061
H2AFX	1.04	0.009827979	H2A histone family member X	ENSG00000188486
HIST1H3A	1.02	0.05944192	histone cluster 1 H3 family member a	ENSG00000275714
HIST2H2BE	0.88	0.028428384	histone cluster 2 H2B family member e	ENSG00000184678
H2AFZ	0.63	0.044623207	H2A histone family member Z	ENSG00000164032
HIST1H3J	0.93	0.204175587	histone cluster 1 H3 family member j	ENSG00000197153
HIST1H4I	0.72	0.065534653	histone cluster 1 H4 family member i	ENSG00000276180
HIST1H2AC	0.47	0.318228769	histone cluster 1 H2A family member c	ENSG00000180573

**Table 2 cells-09-01845-t002:** Cluster of heat-shock protein DEGs between Myr-treated and untreated CF-patients-derived monocytes.

Symbol	Log2FoldChange	Padj	Description	Ensembl
HSPA6	3.41	2.65 × 10^−9^	heat shock protein family A (Hsp70) member 6	ENSG00000173110
DNAJA4	2.71	2.25 × 10^−16^	DnaJ heat shock protein family (Hsp40) member A4	ENSG00000140403
HSPA1B	2.71	1.19 × 10^−10^	heat shock protein family A (Hsp70) member 1B	ENSG00000204388
HSPA1A	2.64	3.02 × 10^−13^	heat shock protein family A (Hsp70) member 1A	ENSG00000204389
DNAJB1	2.47	8.60 × 10^−13^	DnaJ heat shock protein family (Hsp40) member B1	ENSG00000132002
BAG3	2.14	1.79 × 10^−11^	BCL2 associated athanogene 3	ENSG00000151929
HSPA2	2.00	1.54 × 10^−5^	heat shock protein family A (Hsp70) member 2	ENSG00000126803
SERPINH1	1.94	3.78 × 10^−9^	serpin family H member 1	ENSG00000149257
HSPD1	1.68	3.78 × 10^−9^	heat shock protein family D (Hsp60) member 1	ENSG00000144381
ZFAND2A	1.60	1.29 × 10^−9^	zinc finger AN1-type containing 2A	ENSG00000178381
HSPE1	1.40	7.08 × 10^−5^	heat shock protein family E (Hsp10) member 1	ENSG00000115541
HSP90AA1	1.35	1.13 × 10^−6^	heat shock protein 90 alpha family class A member 1	ENSG00000080824
DNAJB4	1.13	5.80 × 10^−7^	DnaJ heat shock protein family (Hsp40) member B4	ENSG00000162616
HSPA1L	1.13	8.07 × 10^−5^	heat shock protein family A (Hsp70) member 1 like	ENSG00000204390
DNAJA1	0.90	6.78 × 10^−5^	DnaJ heat shock protein family (Hsp40) member A1	ENSG00000086061
HSP90AB1	0.82	0.002280106	heat shock protein 90 alpha family class B member 1	ENSG00000096384
HSPA8	0.53	0.058998158	heat shock protein family A (Hsp70) member 8	ENSG00000109971

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
