# Peer review of "Cystic Fibrosis Defective Response to Infection Involves Autophagy and Lipid Metabolism"

_cells, 2020, doi:10.3390/cells9081845_

Round 1

Reviewer 1 Report

In their study, Mingione et al. have analyzed the transcriptome in control and cystic fibrosis (CF) bronchial epithelial cells infected with A. fumigatus. They have also assessed the effects of modulating the sphingolipid pathway using the SPT inhibitor myriocin (Myr). This study is a continuation of prior work from this group which showed that Myr treatment in a CF mouse model reduces inflammation and improves the response to pathogen infection. They find that infected CF and CTRL fibroblasts have distinct transcriptional profiles, and that Myr seems to lead to a more enhanced transcriptional response. Ontology analysis suggests that Myr upregulates autophagy genes and downregulates inflammatory genes. In independent analysis, they validate effects on some inflammatory genes. Using primary CF monocytes treated with and without myriocin, they also identify a cluster of heat shock proteins upregulated on Myr treatment and increased LC3B which they infer as meaning an increase in autophagy/chaperone-induced autophagy.

Overall, this study represents a worthwhile continuation of the prior work – using broad analysis as a way to gain mechanistic insight. Unfortunately, as it stands, the organization of the manuscript and the sheer amount of high throughput data that is given to the reader means that the central message of the study is lost. This should be rectified before the ms is ready for publication. The authors may wish to consider the following:

  1. Given this is a high throughput analysis paper, it would seem more logical to start with the RNAseq data first, and then highlight effects on specific genes (Fig. 1) as validating major findings from the RNAseq analysis. The only real biology I would move higher up is the current Fig 7 showing that Myriocin treatment does modestly increase killing ability. This would better frame the paper as a discovery paper aiming to uncover cell intrinsic mechanisms to account for prior reduced inflammation/increasing killing etc etc. which then leads into RNAseq. Validation of subsequent findings should come at the end. This would create a stronger throughline for the reader to follow.
  2. Along these lines, identifying some of the other major changes in the RNAseq data and validating with independent primers would strengthen the study. If this could be done from the patient derived monocytes, even better.
  3. In the current figure 1, it looks like all data is expressed as fold change over control. A problem with this is that the control bar is now ‘set’ to 1 with no standard error – which will skew the statistical analysis such that changes which may not be statistically significant will now appear so. Along these lines, it looks from the way the graphs are organized that the authors performed a 1 way ANOVA – but the correct test here would be a 2-way ANOVA. The authors should clarify in the statistical section.
  4. Figure 3A and 3B – in the text, the authors talk about profound differences in gene upregulation etc – sometimes referring across the graphs. However, with the different scales on the graphs, it is hard to get an appreciation of the differences. It may be useful to graph the data on the same scales to allow an easier comparison.

Author Response

Reply to Reviewer 1 

Comments

  1. Given this is a high throughput analysis paper, it would seem more logical to start with the RNAseq data first, and then highlight effects on specific genes (Fig. 1) as validating major findings from the RNAseq analysis. The only real biology I would move higher up is the current Fig 7 showing that Myriocin treatment does modestly increase killing ability. This would better frame the paper as a discovery paper aiming to uncover cell intrinsic mechanisms to account for prior reduced inflammation/increasing killing etc etc. which then leads into RNAseq. Validation of subsequent findings should come at the end. This would create a stronger throughline for the reader to follow.

We thank the reviewer for this insightful comment. As suggested, we now open the manuscript with patients’ derived monocytes infection and treatment (Fig 7 is now Fig 1). Next, we analyze the transcriptional effect of Myr on infected monocytes by RNA sequencing. Cell line-based experiments follow, showing the RNA sequencing analysis of CTR and CF cells in infected and infected/Myr treated conditions. Moreover we better specify Myr treatments in the different experimental settings.

  Along these lines, identifying some of the other major changes in the RNAseq data and validating with independent primers would strengthen the study. If this could be done from the patient derived monocytes, even better.

Following the reviewer’s suggestion, we have now added an additional figure showing the RT-PCR validation for the main upregulated genes that have been discussed in the manuscript (Fig.7 of the revised version). Concerning patient derived monocytes, unfortunately, after the RNA-sequencing analysis, we have no RNA left for testing other genes. Indeed, due to Ethical Committee restriction, we obtained a small aliquot of patients’ blood and we managed to obtain cells to prepare replicate samples for infection studies and for RNA-sequencing, but we had no spare biological material to store. Nonetheless, we are truly confident of our results, which have been obtained by differentially expressed gene list, reaching a good p-value despite the high variability among patients. Furthermore, this issue has now been fully discussed in a new paragraph entitled “Strength and limitations of the study”, according to the Reviewer 2 suggestion.

 In the current figure 1, it looks like all data is expressed as fold change over control. A problem with this is that the control bar is now ‘set’ to 1 with no standard error – which will skew the statistical analysis such that changes which may not be statistically significant will now appear so.

The absence of the error bar in controls of qRT-PCR is due to the fact that we are presenting fold increases versus control. The DDCt  evaluation, which is a widely accepted method to evaluate RT-PCR data, requires to use the control as a reference value. Anyway, we always obtained comparable CT among the control samples. We now added this information in the methods section and we thank the Reviewer for pointing out this missing data. Paragraph 2.5. qRT-PCR: “The analysis was performed by referring to control values that do not significantly differentiate (triplicate samples, and their standard deviation value divided by their mean value is <1).”

 Along these lines, it looks from the way the graphs are organized that the authors performed a 1-way ANOVA – but the correct test here would be a 2-way ANOVA. The authors should clarify in the statistical section.

    We thank the Reviewer for this suggestion. We newly performed the statistical analysis by using 2-way ANOVA.

    Figure 3A and 3B – in the text, the authors talk about profound differences in gene upregulation etc – sometimes referring across the graphs. However, with the different scales on the graphs, it is hard to get an appreciation of the differences. It may be useful to graph the data on the same scales to allow an easier comparison.

    We agree with the reviewer that when possible using the same scale is recommended to compare values, but we had highly variable values for the different infected cells versus uninfected, and we have modulated the scales in order to appreciate all differences. To address this issue, we now used logarithmic scales that make appreciable all the differences while keeping a homogeneous reference scale system.

There are minor grammatical errors in several places throughout the manuscript.  Suggest careful English editing prior to re-submission.

This issue has been addressed in the revised manuscript.

We sincerely thank the Reviewer for the very useful remarks that helped us to improve our manuscript. We hope that these additional clarifications and modifications make our paper suitable for publication.

Best regards,

Paola Signorelli

Submission Date

26th June 2020

Date of this review

30th  July 2020 17:16:41

Reviewer 2 Report

This is an interesting study looking at transcriptional responses during aspergillus infection and treatment with Myr in CF epithelial cell lines and monocytes.  Overall, there is useful information to be gained from the study, but clarifications are needed as below.

There are minor grammatical errors in several places throughout the manuscript.  Suggest careful English editing prior to re-submission.

Major:

-There are several methodologic concerns that probably cannot be corrected now but should be noted as limitations of the study.

1) Epithelial Cell lines only were used, not primary epithelial cells (so transcriptional responses may be changed in immortalized cell lines).

2) an aspergillus reference strain (non-CF) was used.  This may not reflect typical responses in CF patients.

3) Myr pre-treatment only was used, it is unclear what would happen if this was given to patients with established infections.

4) Frozen pbmcs were used, this could impact transcriptional responses compared to freshly isolated PBMCs.

5)  only 1 housekeeping gene was used for normalization of data.  It is ideal to use at least 3 housekeeping genes and choose the most stable one. GAPDH is not always stable in CF.

Other issues:

-Figure 1: Non-CF controls should be included for comparison

-figure 3 should include a MyrAsp Control comparison

-Figure 7 – Visual representation of the data is not consistent with the statistical result reported. There is literally only 1 CF treated data point higher than the untreated.   How can this possibly be significant?  the corresponding discussion point should also be rectified if this is truly not significant.

Minor:

  • I don’t understand the difference between the following statements, they seem repetitive. “There is a striking difference in Myr effect in uninfected versus infected cells. We observed a significantly higher number of DEGs in Myr-treated CF (MyrCF, 1629 DEGs) than in CTRL cells (MyrCTRL 55 DEGs) (Figure 3A). Indeed, Myr treatment induces a stronger transcriptome modulation in CF compared to CTRL cells, even in basal conditions (Figure 3A).”
  • Results text for 3.4 says a p value <0.05, but methods state <0.01 for epithelial cells. please clarify
  • Why was the p value set at a different value between epithelial cells and monocytes?
  • Why were non-CF monocytes included as a control in figure 5?
  • Figure 6B expression legend missing the colors to tell which is up or down-regulated
  • A supplemental venn diagram showing overlap between monocytes and epithelial cells would be useful to see what genes are conserved for all cell types in CF
  • Table 2 is referred in the text before table 1
  • Figure 6 legend should denote monocytes used
  • I don’t think the following statement is correct, there are numerous transcriptional studies in CF that have shown similar concepts, please rectify. ‘To the best of our knowledge, this is the first clear evidence that a profound modification in gene expression, triggered by infection, differentiates CF from healthy cells, suggesting that the disease itself causes higher sensitivity to infection.

Author Response

Reply to Reviewer 2

Comments

Major:-There are several methodologic concerns that probably cannot be corrected now but should be noted as limitations of the study.                                                                                                      1) Epithelial Cell lines only were used, not primary epithelial cells (so transcriptional responses may be changed in immortalized cell lines).                                                                                                        2)  an aspergillus reference strain (non-CF) was used.  This may not reflect typical responses in CF patients.                                                                                                                                                         3) Myr pre-treatment only was used, it is unclear what would happen if this was given to patients with established infections.                                                                                                                                4) Frozen pbmcs were used, this could impact transcriptional responses compared to freshly isolated PBMCs.                                                                                                                                                     5)  only 1 housekeeping gene was used for normalization of data.  It is ideal to use at least 3 housekeeping genes and choose the most stable one. GAPDH is not always stable in CF.

We sincerely thank the Reviewer for pointing out such important issues and we have now included an additional section entitled “Strength and limitations of the study” at the end of the discussion. We believe that this section may contribute to a fair evaluation of the data from the Readers and may help the development of further research in the field.

Other issues:

-Figure 1: Non-CF controls should be included for comparison

According to Reviewer 1 suggestions, we revised data presentation, and figure 1 is now figure 8. At first, we did not include control cells in the RT-PCR data as, by applying the DDCt methods, control cells are used to normalize data (calibrator).  We have now added this data (the basal expression levels of studied genes in CTRL and CF cell lines) in Supplementary Figure 1.

-figure 3 should include a MyrAsp Control comparison

Figure 3 is now named Figure 5.This figure depicts basal differences in gene expression between control and CF cells (panel A), and Myr modulation of such a response during infection with A. fumigatus (panel B).Therefore in panel B it is possible to compare MyrAsp CF and MyrAsp CTRL. The Venn diagram (panel C) is intended to highlight how Myr treatment, resulting in a higher number of genes commonly regulated within the two cell lines, modulated CF phenotype slightly towards the normal phenotype, upon infection. To simplify data interpretation, we omitted MyrAsp Control in the diagram.

 -Figure 7 – Visual representation of the data is not consistent with the statistical result reported. There is literally only 1 CF treated data point higher than the untreated.   How can this possibly be significant?  the corresponding discussion point should also be rectified if this is truly not significant.

We amended the figure providing a column bar graph (using mean with 95% CI). We hope this format to be more representative of the obtained statistical analysis (p= 0.0013). In particular, we performed Wilcoxon matched-pairs signed rank test, considering the killing ability of patient-derived monocytes before and after Myr treatment. 

 I don’t understand the difference between the following statements, they seem repetitive. “There is a striking difference in Myr effect in uninfected versus infected cells. We observed a significantly higher number of DEGs in Myr-treated CF (MyrCF, 1629 DEGs) than in CTRL cells (MyrCTRL 55 DEGs) (Figure 3A). Indeed, Myr treatment induces a stronger transcriptome modulation in CF compared to CTRL cells, even in basal conditions (Figure 3A).”

We thank the reviewer for highlighting this redundancy. The paragraph has been amended in: “We observed a significantly higher number of DEGs in Myr-treated CF (MyrCF, 1629 DEGs) than in CTRL cells (MyrCTRL 55 DEGs) (Figure 5 A)”.

  • Results text for 3.4 says a p-value <0.05, but methods state <0.01 for epithelial cells. please clarify
  • Why was the p value set at a different value between epithelial cells and monocytes?

Regarding the cell lines experiment, we set an adjusted p-value of 0.01 in order to be more stringent on DEGs definition due to the low number of replicates (3 vs 3 comparisons). Moreover, downstream analyses for the cell line experiment are based on GSEA, in which we used all the genes pre-ranked for logFC values. Hence, GSEA results are not affected by DEGs number. The different p values are specified in line 189-190.

  • Why were non-CF monocytes included as a control in figure 5?

Figure 5 is now named figure 2 in the revised manuscript. We did not include healthy monocytes in this project (Ethical Committee Approval) as the aim of our study was to evaluate MYR effects in A. fumigatus-infected CF monocytes (vs. infected and untreated).

  • Figure 6B expression legend missing the colors to tell which is up or down-regulated

This is now Figure 3B. We thank the reviewer for highlighting this missing information. In the revised manuscript, we amended the figure legend.

  • A supplemental venn diagram showing overlap between monocytes and epithelial cells would be useful to see what genes are conserved for all cell types in CF

We thank the Reviewer for this valuable suggestion and we now introduced a new Venn diagram to trace a possible overlap in Myr induced genes in epithelial versus monocytes CF cells, infected with A.fumigatus. This new figure significantly strengthens the message of our manuscript because of the role of such few genes in the highlighted pathways.

Table 2 is referred in the text before table 1

We thank the reviewer for let us notice the oversight. In the revised manuscript, Supplementary Table 1 is mentioned in the Methods Section and  Table 1 and Table 2 are mentioned in the paragraph 3.1, consecutively.

Figure 6 legend should denote monocytes used

This issue has been addressed

  • I don’t think the following statement is correct, there are numerous transcriptional studies in CF that have shown similar concepts, please rectify. ‘To the best of our knowledge, this is the first clear evidence that a profound modification in gene expression, triggered by infection, differentiates CF from healthy cells, suggesting that the disease itself causes higher sensitivity to infection.

We agree with the Reviewer and the sentence has been appropriately modified in the discussion.

We sincerely thank the Reviewer for the very useful remarks that helped us to improve our manuscript. We hope that these additional clarifications and modifications make our paper suitable for publication.

Best regards,

Paola Signorelli

Submission Date

26th June 2020

Date of this review

30th  July 2020 17:16:41

Round 2

Reviewer 1 Report

The authors had addressed the previous critiques sufficiently and the manuscript is much improved. It is now acceptable for publication.